# AdaptInfer: Adaptive Token Pruning for Vision–Language Model Inference with Dynamical Text Guidance

## Abstract

Vision–language models (VLMs) have achieved impressive performance on multimodal reasoning tasks such as visual question answering, image captioning and so on, but their inference cost remains a significant challenge due to the large number of vision tokens processed during the prefill stage. Existing pruning methods often rely on directly using the attention patterns or static text prompt guidance, failing to exploit the dynamic internal signals generated during inference. To address these issues, we propose AdaptInfer, a plug-and-play framework for adaptive vision token pruning in VLMs. First, we introduce a fine-grained, dynamic text-guided pruning mechanism that reuses layer-wise text-to-text attention maps to construct soft priors over text-token importance, allowing more informed scoring of vision tokens at each stage. Second, we perform an offline analysis of cross-modal attention shifts and identify consistent inflection locations in inference, which inspire us to propose a more principled and efficient pruning schedule. Our method is lightweight and plug-and-play, also generalizable across multimodal tasks. Experimental results have verified the effectiveness of the proposed method. For example, it reduces CUDA latency by 61.3% while maintaining an average accuracy of 93.1% on vanilla LLaVA-1.5-7B. Under the same token budget, AdaptInfer surpasses SOTA in accuracy.

## 1 Introduction

In recent years, building on the success of LLMs (Bommasani et al., 2021; Touvron et al., 2023; Brown et al., 2020), vision–language models (VLMs) have emerged to tackle multimodal reasoning by combining visual encoders  (Liu et al., 2022; Dosovitskiy et al., 2021), with LLMs' text decoders (Du et al., 2022). This integration enables impressive performance on tasks such as captioning (Lin et al., 2014), image retrieval (Faghri et al., 2018), and visual question answering (VQA) (Antol et al., 2015), but it also introduces a new computational challenge: the sheer number of vision tokens.

During the inference process of VLMs, the number of vision tokens is often much larger than that of textual tokens, sometimes by an order of magnitude or more. For instance, an image of size $672 \times 672$ processed by a visual encoder with a patch size of $14 \times 14$ typically results in 2304 vision tokens (Radford et al., 2021; Zhang et al., 2025), whereas the corresponding text prompt may contain fewer than 100 tokens (Hudson & Manning, 2019; Fu et al., 2023). Also, much previous research suggests that the vision tokens are more redundant and semantically repetitive (Zhang et al., 2024; Tong et al., 2025).

As a result, explorations in VLM acceleration primarily focus on the efficient pruning, compression or sparsification of vision tokens. This paradigm aims to reduce computational overhead by retaining only the most valuable vision tokens. Among them, some works introduce sparsity strategies within the visual encoders to generate less but useful enough vision tokens (Li et al., 2024; Chen et al., 2025b), while others further prune tokens based on either self-attention or cross-attention patterns during the prefill stage (Lin et al., 2025; Xing et al., 2024; Chen et al., 2025a; Li et al., 2025). Nevertheless, not all the attention logits should be involved in the vision token ranking for the

dispersion of the full attention patterns (Zhang et al., 2024). Granting voting rights only to the most salient tokens sharpens guidance, enabling more aggressive yet accurate vision-token pruning.

To address this, SparseVLM (Zhang et al., 2025) introduces the concept of text prompt–guided pruning, selecting the most salient text tokens offline before the prefill pass. While this approach acknowledges the importance of textual cues, it does not fully address the underlying challenge: **the dynamic nature of text token importance during inference**. In practice, the informativeness of text tokens evolves across layers as the model progressively refines its internal representations (Tenney et al., 2019; Clark et al., 2019). Our observations in Figure 1a also indicate that the most prominent text tokens vary significantly across layers, making any static selection inherently suboptimal.

Therefore, to truly harness the benefits of text-guided sparsification, it is crucial to develop a pruning strategy matched with dynamic fluidity of information, reflecting the effective cross-modal interaction throughout the inference process. In this work, we propose to reconstruct the dynamic importance ranking of text tokens at each layer by utilizing the text-to-text (t2t) attention maps. These attention maps provide a natural, layer-specific prior distribution for text-token importance, which we then use to reweight text-to-vision (t2v) attention scores for vision token pruning. Importantly, since the t2t attention maps can be directly extracted from the model's attention computations, our method does not introduce additional computational overhead.

Moreover, current methods determine pruning hyperparameters (e.g. the pruning locations) primarily through either empirical rule-of-thumb or extensive hyperparameter optimization experiments (Xing et al., 2024; Chen et al., 2025a; Zhang et al., 2024). However, we argue that **relying on manual tuning or grid search not only imposes substantial offline computational overhead, but also leads to task- or dataset-specific heuristics**. In this work, we take the first step in providing a principled pruning schedule. We provide our insights based on systematically analyzing the distributional characteristics of attention shifts to vision tokens during VLM inference. Specifically, we identify consistent attention inflection points at layer 1, 10, and 20 in LLava-1.5-7B (Liu et al., 2023b), suggesting that aggressive pruning immediately after these layers is a more effective and computationally efficient strategy on LLava.

The solutions we propose effectively address the limitations of existing works in the field of vision token sparsification for VLM acceleration. Our main contribution involves:

- We propose AdaptInfer, an adaptive vision token sparsification framework in which VLM dynamically determines text token guidance during inference. AdaptInfer is a plug-and-play solution.
- We introduce a novel observation of the distributional characteristics of attention shifts, and gain insights in a more effective and reasonable pruning schedule.
- We implement and evaluate our proposed solution, AdaptInfer, across multiple benchmarks and different vision token budget settings. Within the same token budget, our AdaptInfer outperforms state-of-the-art (SOTA) methods on the metric of the accuracy.

## 2 RELATED WORK

In this section, we will briefly introduce the previous works that are correlated with ours.

### 2.1 VISION-LANGUAGE MODELS

Early multi-modal systems paired convolutional vision backbones with recurrent language decoders (Karpathy & Fei-Fei, 2017; Vinyals et al., 2015). Modern VLMs instead follow the Transformer paradigm (Vaswani et al., 2017) by representing an image as a sequence of *visual tokens* that interact with textual tokens in a shared self-attention space (Liu et al., 2023b; Chen et al., 2024a;b). BLIP-2 (Li et al., 2023a) and MiniGPT-4 (Zhu et al., 2023) introduce lightweight linear adapters that project features from a frozen CLIP encoder into the hidden space of a large language model, enabling efficient training. These explorations bridge the gap between frozen encoders and LLMs. The LLaVA family (Liu et al., 2023a;b; 2024a) refines this recipe with stronger instruction tuning, while a few other efforts such as Flamingo (Alayrac et al., 2022), CogVLM (Wang et al., 2024b) and GPT-4V (OpenAI, 2023) scale the approach to billions of parameters.

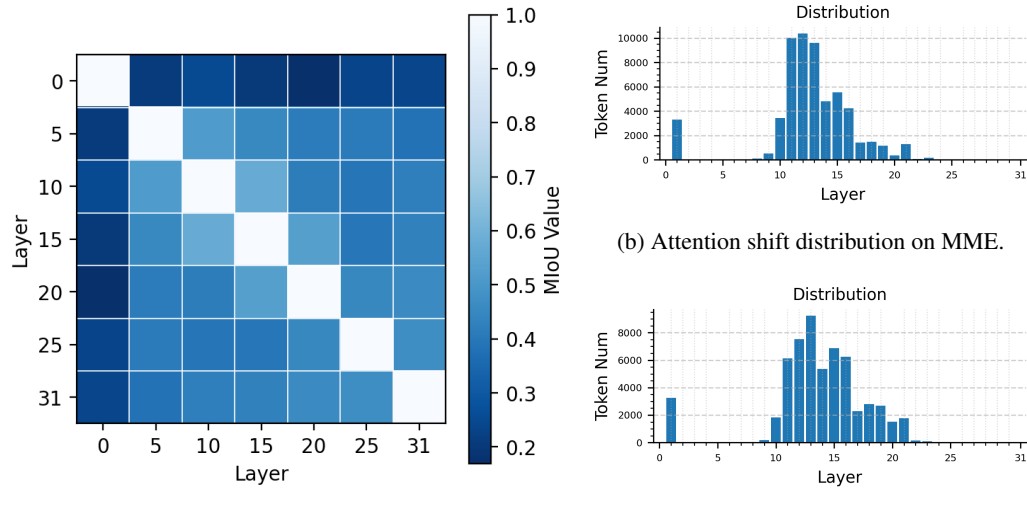

(a) MIoU of key text tokens in different layers.

(b) Attention shift distribution on MME.

(c) Attention shift distribution on TextVQA.

Figure 1: **Preliminary observations.** (a) The consistent low mIoU shows the key text token varies across layers. (b)(c) Layer-wise distribution of attention shifts on MME and TextVQA, which shows a highly consistent trend.

Recent explorations also contribute to resolution and multi-modal extensions. Models leveraging hierarchical perception (e.g., LLaVA-NeXT (Liu et al., 2024a)) and adaptive patching (e.g., Qwen-VL (Bai et al., 2023)) permit high-resolution inputs, while large-scale vision encoders are adopted to generate richer hidden states of the vision tokens (Chen et al., 2024a;b). However, these gains come at the cost of dramatically more vision tokens, which is a key bottleneck addressed by our work.

## 2.2 INFERENCE ACCELERATION OF VLMS

Previous approaches mainly focus on vision token sparsification. This is because the number of vision tokens is often an order of magnitude (or more) larger than that of text tokens. In addition, visual embeddings are naturally much more sparse and repetitive than human-made texts (Marr, 2010). In this field, there are two research directions including efficient vision encoders and vision token pruning in LLM networks.

For example, methods like LLaVA-PruMerge (Shang et al., 2025) and FlowCut (Tong et al., 2025) follow the first direction, which cuts the encoder outputs or uses a lightweight projector to reduce the number of vision tokens. Recoverable compression (Chen et al., 2025b) is also introduced to repair the information loss from token pruning within vision encoders. Solutions follow the second direction not only to drop the vision tokens (Chen et al., 2025a; Lin et al., 2025; Xing et al., 2024; Li et al., 2025; Luan et al., 2025; Ye et al., 2025) during prefill, but also merge them to compress the numbers of the vision tokens (Bolya et al., 2023) and recover them in certain inference stage (Wu et al., 2025). SparseVLM (Zhang et al., 2025) tries to go deeper in exploration of static text prompt guidance but ignoring the evolving inherent of token information. Our approach contributes to the second paradigm.

## 3 METHOD

In this section, we will introduce our observations and the methods we propose.

## 3.1 OBSERVATIONS

In this subsection, we will present a few observations and preliminary experiments that have illuminated our insights.

### 3.1.1 TEXT TOKEN IMPORTANCE

Given an input question prompt, we first tokenize it as $Prompt = [t_1, \ldots, t_n]$. We then define the *importance* of a text token $t_i$ at layer $\ell$ as the total attention weight it receives from all text tokens in the layer-$\ell$ text-to-text (t2t) attention map on average of attention heads:

$$\text{Imp}_\ell(t_i) = \frac{1}{H} \sum_{h=1}^{H} \sum_{j=1}^{n} \mathbf{A}_{\text{t2t}}^{(\ell,h)}[j, i], \tag{1}$$

where $\mathbf{A}_{\text{t2t}}^{(\ell,h)}[j, i]$ denotes the attention of head $h$ directed from token $t_j$ to $t_i$, $H$ represents the number of the attention heads. Intuitively, tokens with high $\text{Imp}_\ell$ are the current key text tokens of the language stream and are therefore best suited to guide cross-modal pruning decisions at that layer.

### 3.1.2 DYNAMICS OF TEXT TOKEN IMPORTANCE

To demonstrate that the importance of text tokens evolves significantly across layers, we conduct a simple empirical study on LLava-1.5-7B (Liu et al., 2023b). LLava-1.5-7B contains a 32-layer LLaMa (Touvron et al., 2023) as its language model. We extract t2t attention maps from 1,000 samples in the TextVQA dataset. On average, each sample contains approximately 100 text tokens. At each chosen layer, we select the top 20% text tokens as the key text tokens of that layer. We then compute the mean Intersection over Union (mIoU) between the indexes of the selected top tokens across layers, as reported in Figure 1a. The value in the i-th row and j-th column represents the mIoU of the key text token indexes of the i-th and j-th layers. Note that this figure is symmetrical, and the miou values along the diagonal are always 1 because they are comparisons within the same layer.

The consistently low MIoU values between different layers indicate that the set of the key text tokens changes substantially during inference. For example, the 0.169 mIoU between layer 0 and 24 refers that only 16.9% of key text tokens are overlapped while all others are different. This highlights the inherent dynamics of text token importance, where the VLM attends to different parts of the input question at different stages of reasoning. Consequently, any static text-prompt–guided pruning approach is likely to fail in capturing this evolving semantic alignment. These findings support the need for an adaptive, layer-wise text-guidance mechanism that can track and respond to the evolving attention distribution online.

### 3.1.3 CROSS-ATTENTION SHIFTS OF VLM

We argue that a principled approach to setting pruning hyperparameters is not only necessary but preferable to complex, trial-and-error–based tuning. To support this claim, we performed an analysis to investigate the locations of cross-attention shifts during inference. Specifically, we calculate cumulative text-to-vision (t2v) attention scores for visual tokens at each layer using LLava-1.5-7B (Liu et al., 2023b). For each sample, we first select the top 10% of vision tokens, approximately 58 tokens per image, which receive the highest total attentions in the prefill stage. Similar to the text tokens, these vision tokens are assumed to be the most critical ones for the corresponding image-based multi-modal task.

To understand when these important tokens become semantically salient, we analyze their cumulative attention trajectories across transformer layers. We apply change-point detection (Truong et al., 2020) on each curve to identify the layer where the model's attention pattern changes significantly. The technical details of change-point detection are presented in the Appendix B. Intuitively, an attention shift point may indicate that either (1) the token begins to receive significantly more attention, becoming critical, or (2) the token's informative content has already been fully extracted, becoming redundant. In both cases, these shifts mark the layer-wise transitions in how the model utilizes visual information.

Figure 1b and 1c shows the distribution of the attention shift locations aggregated from 1,000 samples each in the MME (Fu et al., 2023) and TextVQA (Singh et al., 2019) datasets. Despite data set differences, we observe a highly consistent trend: attention shifts cluster densely in layer 1 and round layers 10-20, while layers 2-9 and 20 + show a frequency of low attention shifts. These find-

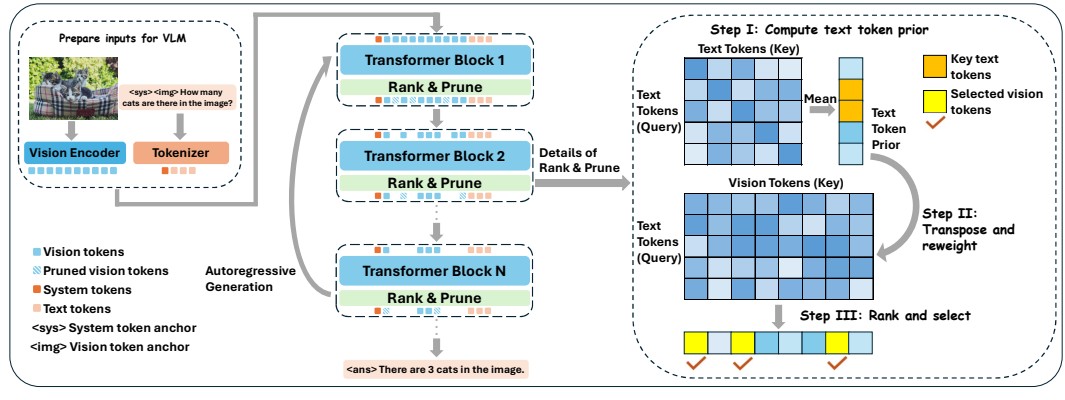

Figure 2: **The architecture of AdaptInfer**. Text token importance is computed and guides vision token selection at every pruning layer adaptively. The right panel illustrates the internal computation details of the Rank & Prune module. *Step I*: Compute the text token prior. *Step II*: Use the transposed prior to reweight the t2v attention. *Step III*: Rank the vision tokens and prune the least informative ones.

ings provide a data-driven basis for pruning schedule design, where pruning locations are informed by the model's own attention behavior, rather than by empirical intuition alone.

### 3.2 ADAPTIVE TOKEN PRUNING

Based on the two key observations above, we propose AdaptInfer for VLM inference acceleration.

#### 3.2.1 DYNAMIC TEXT GUIDANCE PRUNING

To guide visual token pruning in a more informed and adaptive way, we propose a dynamic text-guidance mechanism that leverages the model's internal attention signals. Instead of statically selecting a fixed subset of text tokens before entering the language model (Zhang et al., 2025), we dynamically infer their relative importance during inference at each predefined pruning layer. We reuse the attention maps already computed by the model. The architecture of Our AdaptInfer is shown in Fig 2, and our dynamic text guidance mechanism can be divided into three steps.

Firstly, in each pruning layer, we extract the *text-to-text* attention matrix $\mathbf{A}_{t2t}^h \in \mathbb{R}^{T \times T}$, where $T$ is the number of text tokens, $h$ is the index of attention head with a total number of $H$. To estimate the importance of each text token, we aggregate attention scores across the query dimension:

$$\mathbf{w}^{(\mathbf{h})} = \sum_{i=1}^{T} \mathbf{A}_{t2t}^{(h)}[i, :] \in \mathbb{R}^T, \tag{2}$$

where $\mathbf{w}$ serves as a soft prior distribution over the text tokens, averaged across all attention heads, indicating how much attention each token receives from the rest of the sequence.

Secondly, we then use this prior to reweight the t2v attention matrix $\mathbf{A}_{t2v}^{(h)} \in \mathbb{R}^{T \times V}$, where $V$ is the number of visual tokens remained. $\mathbf{A}_{t2v}^{(h)}$ denotes the cross-attention matrix where text tokens are used as queries, and vision tokens serve as keys and values. The importance score of each visual token on average of all attention heads is computed as:

$$\mathbf{s} = \frac{1}{H} \sum_{h=1}^{H} \mathbf{w}^{(h)\top} \cdot \mathbf{A}_{t2v}^{(h)} \in \mathbb{R}^V. \tag{3}$$

Here, $\mathbf{s}_j$ reflects the aggregated and weighted attention from all text tokens to visual token $j$.

Finally, based on these scores, we rank all visual tokens and retain the top-$k$ for the current layer:

$$\mathcal{I}_k = \text{TopK}(\mathbf{s}, k). \tag{4}$$

Importantly, all text tokens participate in visual token scoring, but contribute in proportion to their dynamically inferred importance. Moreover, because both $\mathbf{A}_{t2t}$ and $\mathbf{A}_{t2v}$ are natively computed in standard forward passes, our method introduces little additional computational overhead. Note that this solution follows a training-free paradigm and can be seamlessly integrated as a plugin into existing VLMs.

### 3.2.2 ANALYSIS OF COMPUTATIONAL COMPLEXITY

Assuming $n = T + V$ denotes the current sequence length, $d$ denotes the VLM's hidden state dimension, and $m$ denotes the hidden size of projection layer in the FFN network. For each transformer layer in the prefill stage, the FLOPs can be estimated by

$$FLOPs^{\text{prefill}} = 4nd^2 + 2n^2d + 3ndm. \tag{5}$$

For each pruning layer, the additional FLOPs of are computed below:

$$FLOPs^{\text{prune}} = T^2 + 2TV. \tag{6}$$

Since both attention matrices are already computed during the forward pass, this additional cost is minimal relative to the main transformer computations. Then, during the decode stage, the FLOPs of each layer can be estimated by

$$FLOPs^{\text{decode}} = 4d^2 + 2nd + 3dm. \tag{7}$$

### 3.2.3 LAYER-WISE PRUNING SCHEDULE

Following the attention shift analysis described above, we design a pruning schedule that aligns with the attention dynamics observed in VLMs. Pruning visual tokens introduces an inherent trade-off between pruning safety and computational savings. To prune aggressively, one must prune early; yet pruning too early inevitably risks removing tokens whose importance has not yet emerged.

Prior works rely directly on attention scores to rank tokens (Xing et al., 2024; Zhang et al., 2025; Chen et al., 2025a). However, we argue that attention scores from all transformer layers are not equally reliable for serving as evidence for pruning. Our attention shift can indicate how reliable the attention-based rankings are across layers. Our attention-shift analysis helps to reveal two schemes: Firstly, in the high-frequency regions (e.g., layer 1 and layers 10–20), token importance is actively being reassigned (either becoming important for the first time or being used up by the model). Attention rankings here are unstable and thus unsafe for pruning. Secondly, in the stable regions (e.g., layers 2–9 and 20+), importance rankings remain consistent and therefore provide reliable pruning signals.

Based on the discussion above, we select to prune vision tokens after layer 1 and after layer 20 on Llava-1.5-7B. Both locations mark the beginning of each stable region, enabling early pruning to save more FLOPs relying on stable attention rankings. To prevent over-pruning at layer 1, which would remove too many informative tokens too early, we choose an additional pruning location between layers 1 and 20. This step is roughly to be placed at layer 10, the midway of layer 1 and 20, which ensure that (1) sufficient depth for each pruning to take effect and (2) little interference with the high-volatility band. This schedule balances caution and efficiency. Compared to heuristic or uniform pruning schemes, our approach is both data-driven and architecture-aware, requiring no expensive hyperparameter tuning while generalizing well across tasks and datasets.

### 3.3 DISCUSSION

The chosen pruning hyperparameters come from empirical observations on LLaVA-1.5-7B, and thus are tailored specifically to VLMs built upon the LLaMA-7B backbone. Nevertheless, our proposed adaptive pruning schedule is generalizable for it can be easily transferred to other models with different parameter scales or architectures by performing a simple, offline attention shift analysis. Additional experiments on LLaVA-1.5-13B and Qwen2-VL-2B (Wang et al., 2024a) are presented in

the Appendix C and D. Based on the attention shift distrubations, we select pruning locations at layer 1, 11 and 22 on LLaVA-1.5-13B, and at layer 0, 9 and 19 on Qwen2-VL-2B for later experiments.

Moreover, one of the datasets used in our observational studies is MME (Fu et al., 2023), a comprehensive multimodal benchmark comprising two major categories and fourteen subcategories. The consistent statistical patterns indicate that the attention dynamics are largely stable and transferable across different types of multimodal tasks.

## 4 Experiment

### 4.1 Experimental Settings

#### 4.1.1 Datasets

For multimodal evaluation, we test our solutions on five widely used benchmarks, including MME (Fu et al., 2023), GQA (Hudson & Manning, 2019), MMBench (MMB) (Liu et al., 2024b), ScienceQA (SQA) (Lu et al., 2022), TextVQA (TVQA) (Singh et al., 2019), and POPE (Li et al., 2023b). These datasets together provide a comprehensive evaluation, including vision-question-answering (VQA), optical character recognition (OCR), perception, reasoning, factual grounding and so on.

In addition, we also test our solutions on three popular video-based multimodal benchmarks, TGIF-QA (TGIF) (Jang et al., 2019), MSVD-QA (MSVD) and MSRVTT-QA (MSRVTT) (Xu et al., 2017).

#### 4.1.2 Baselines

We select four classic and latest vision token sparsification frameworks for VLM acceleration within the plug-and-play paradigm in LLM forward pass as baselines, including FastV (Chen et al., 2025a), ToMe (Bolya et al., 2023), Pyramid Drop (PDrop) (Xing et al., 2024) and SparseVLM (Zhang et al., 2025). For PDrop, we only adopt the training-free version of PDrop to fit our requirements. Note that, methods performing token pruning or merging anywhere other than the prefill stage of language models, like vision encoders or the decode stage (Yang et al., 2024; Tong et al., 2025), are not included for comparison, since such approaches are orthogonal to ours and can be used together.

#### 4.1.3 Implement Details

To ensure a fair and comprehensive comparison between baseline methods and AdaptInfer, we report the results in different average retained token budgets (e.g. 128, 64, 48 and 32). AdaptInfer utilizes the pruning parameters described above while other baselines keep their original settings for all core components. Our experiments are performed on two different types of VLMs, including LLava-1.5-7B (Liu et al., 2023b) and InternVL-chat-7B (Chen et al., 2024a). Additional experiments on LLava-1.5-13B are presented in appendix C.2.

### 4.2 Main Results

In Table 1, we report the performance of AdaptInfer on LLava-1.5-7B. We provide comparisons with other baselines under average retained vision token numbers of 128 and 64. The accuracy scores has an upper bound of Vanilla LLava with all 576 tokens kept, so an average accuracy ratio is supported by each baseline. As shown in this Table, AdaptInfer achieves the highest overall accuracy scores under both 128 and 64 vision token budgets. Under 128 token budgets, our average ratio reaches 97.5%, which is 0.7% higher than the second-best method, SparseVLM. While this may seem like a small margin, it is in fact approaching the upper bound of Vanilla LLava. In practice, retaining only 64 tokens on average reduces the prefill token load by 88.9%, enabling significantly more efficient inference. Despite this aggressive pruning, our framework still preserves 93.1% of the original inference accuracy and outperforms SparseVLM, which scores 91.4%, with a clear improvement of 1.7%. In Figure 3, we present the performance trends of AdaptInfer with the latest baseline methods SparseVLM and PDrop. AdaptInfer outperforms others with clear margins.

We further evaluate the proposed AdaptInfer on InternVL-Chat-7B (Chen et al., 2024a) in Table 2. InternVL shares the same LLaMA architecture but adopts a more powerful 6B parameter ViT en-

Table 1: **Comparison of methods under different pruning budgets on LLava-1.5-7B**. We report results on five datasets, including average retained tokens, accuracy scores and average ratios. Results in bold present the highest accuracy under same pruning budgets.

| Method | Avg. Tokens | MME | GQA | MMB | SQA | TVQA | Ratio (%) |
|---|---|---|---|---|---|---|---|
| Vanilla | 576 | 1864 | 61.9 | 64.6 | 69.5 | 58.3 | 100 |
| ToMe (ICLR23) | 128 | 1343 | 52.4 | 53.3 | 59.6 | 49.1 | 81.8 (↓ 18.2) |
| FastV (ECCV24) | 128 | 1490 | 49.6 | 56.1 | 68.6 | 52.5 | 87.1 (↓ 12.9) |
| PDrop (CVPR25) | 128 | 1761 | 57.1 | 61.6 | 68.4 | 56.6 | 95.5 (↓ 4.5) |
| SparseVLM (ICML25) | 128 | 1746 | 58.4 | **64.5** | 68.6 | 56.7 | 96.8 (↓ 3.2) |
| **AdaptInfer (Ours)** | 128 | **1794** | **58.5** | 63.8 | **69.9** | **56.8** | **97.5 (↓ 2.5)** |
| ToMe (ICLR23) | 64 | 1138 | 48.6 | 43.7 | 50.0 | 45.3 | 71.4 (↓ 28.6) |
| FastV (ECCV24) | 64 | 1255 | 46.1 | 47.2 | 68.7 | 45.9 | 78.5 (↓ 21.5) |
| PDrop (CVPR 25) | 64 | 1561 | 47.5 | 58.8 | 69.0 | 50.6 | 87.5 (↓ 12.5) |
| SparseVLM (ICML 25) | 64 | 1589 | **53.8** | 60.1 | 69.8 | 53.4 | 91.4 (↓ 8.6) |
| **AdaptInfer (Ours)** | 64 | **1684** | 53.2 | **61.7** | **69.9** | **54.3** | **93.1 (↓ 6.9)** |

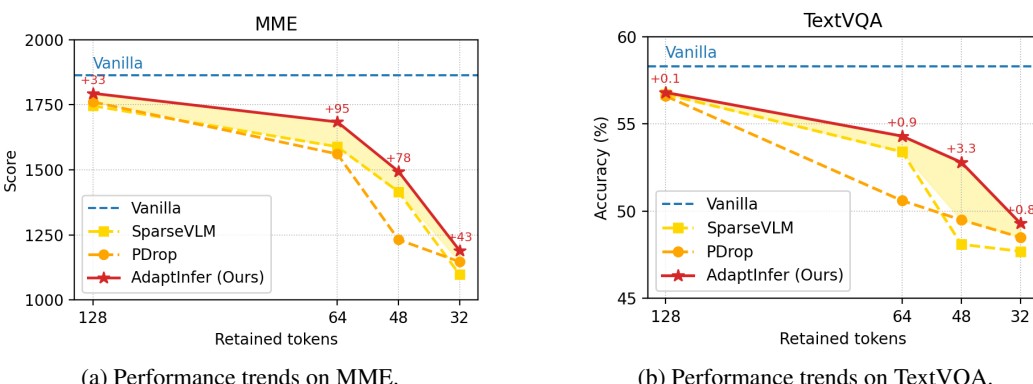

(a) Performance trends on MME.  (b) Performance trends on TextVQA.

Figure 3: **Performance trends on two datasets.** AdaptInfer outperforms SparseVLM and PDrop across all retained token budgets.

coder, which produces finer-grained vision tokens with richer hidden states. Experiments confirm that our token-sparsification strategy preserves the original semantic richness of these vision features with high inference accuracy. Under token budgets of 128 and 64, AdaptInfer maintains a high accuracy ratio of 97.7% and 91.9% respectively. However, in the case of extreme pruning, where only 32 average vision tokens are retained, the accuracy has a significant drop to only 81.5%. This is because the inevitable removal of a large number of informative tokens affects the performance.

## 4.3 EVALUATION ON QWEN2-VL

Qwen2-VL (Wang et al., 2024a) is a more recent and robust VLM series that not only support image-based tasks, but also support video question answering (video QA) tasks. To provide a boarder evaluation, we test AdaptInfer on 8 benchmarks, including 3 video QA datasets (TGIF-QA, MSRVTT-QA, MSVD-QA). Unlike LLaVA, Qwen2-VL dynamically adjusts its vision token counts. Thus, for fairness, we compare SparseVLM and AdaptInfer under the same retained-token ratios (50%, 30%, 10%). The results are summarized in Table 3. Across both image-based QA and multi-frame video QA, AdaptInfer consistently matches or outperforms SparseVLM. Our method performs better particularly under the 30% and 10% retained-token settings, where the pruning becomes more aggressive.

Interestingly, at the 50% retained ratio, both SparseVLM and AdaptInfer slightly outperform the original model. We believe this reflects a widely observed property in multimodal QA: the presence of a large number of redundant visual tokens. Removing these distractors not only accelerates inference but can also improve QA performance by reducing noise. This further confirms the strong

Table 2: **Performance of AdaptInfer on InternVL.** Our method still maintain high accuracy.

| Tokens | MME | SQA | TVQA | Ratio (%) |
|--------|------|------|------|-----------|
| Origin | 1849 | 69.1 | 56.9 | 100 |
| 128 | 1758 | 69.2 | 55.7 | 97.7 |
| 64 | 1528 | 69.0 | 53.0 | 91.9 |
| 32 | 1183 | 68.1 | 46.7 | 81.5 |

practical value of token pruning. More implementation details of AdaptInfer on Qwen2-VL-2B are disclosed in Appendix I.4.

Table 3: **Performance of AdaptInfer on Qwen2-VL-2B.** Result in bold indicate higher.

| Methods | Tokens | Image-Based | | | | | Video-Based | | | Ratio |
|---------|--------|------|------|------|------|------|------|--------|------|-------|
| | | MME | TVQA | GQA | MMB | POPE | TGIF | MSRVTT | MSVD | |
| Vanilla | 100% | 1901 | 77.8 | 60.4 | 71.7 | 86.9 | 9.9 | 30.9 | 42.3 | 100% |
| SparseVLM | 50% | 1900 | **76.9** | 59.9 | 71.0 | 86.4 | **11.7** | **31.2** | 42.9 | **102.1%** |
| **AdaptInfer** | 50% | **1909** | 76.6 | **60.0** | **72.1** | **86.7** | 11.1 | 30.5 | **43.6** | 101.7% |
| SparseVLM | 30% | 1867 | 73.6 | 57.5 | 70.1 | 84.6 | 9.3 | 29.5 | 41.8 | 96.4% |
| **AdaptInfer** | 30% | **1885** | **73.9** | **58.5** | **71.4** | **85.8** | **10.4** | **30.2** | **43.5** | **99.4%** |
| SparseVLM | 10% | 1460 | 51.3 | 40.4 | 57.3 | 40.3 | 5.0 | 28.1 | 30.2 | 68.6% |
| **AdaptInfer** | 10% | **1721** | **62.3** | **48.8** | **63.7** | **71.3** | **5.9** | **28.2** | **41.4** | **81.6%** |

## 4.4 LATENCY TEST

In addition, we conduct a comparative analysis of CUDA latency time, and FLOPs on LLaVA-1.5-7B, in order to show the real acceleration performance. The results are shown in Table 4. This experiment is carried out on a single NVIDIA RTX 4090 24G GPU. All results are the average values per sample.

We replicate the performance of PDrop (Xing et al., 2024) and SparseVLM (Zhang et al., 2025) under 64-token budget, and compare them with that of AdaptInfer. The comparisons are based on four commonly used benchmarks, including MME, GQA, MMBench (MMB) and TextVQA (TVQA). According to the table, while theoretical FLOPs estimations are close, our implementation on AdaptInfer reaches a lower average cuda latency of 33.0 ms per sample than 34.5 ms and 36.7 ms by PDrop and SparseVLM. After all, our method rarely introduces additional computational load, such as any extra attention computation or offline feature matching step.

Table 4: **Latency test of AdaptInfer.** While the FLOPs estimations are close, our method reduces more real CUDA latency.

| Method | Tokens | Metrics | MME | GQA | MMB | TVQA | Average |
|--------|--------|---------|------|------|------|------|---------|
| Vanilla | 576 | FLOPs (T) | 4.268 | 4.250 | 4.623 | 4.611 | 4.438 |
| | | Latency (ms) | 82.0 | 76.5 | 91.3 | 91.2 | 85.3 |
| PDrop | 64 | FLOPs (T) | 0.975 | 0.958 | 1.316 | 1.305 | 1.138 ($\downarrow$ 74.3%) |
| | | Latency (ms) | 33.0 | 32.0 | 36.4 | 36.6 | 34.5 ($\downarrow$ 59.6%) |
| SparseVLM | 64 | FLOPs (T) | 0.974 | 0.958 | 1.316 | 1.305 | 1.138 ($\downarrow$ 74.3%) |
| | | Latency (ms) | 34.4 | 34.7 | 38.1 | 39.5 | 36.7 ($\downarrow$ 57.0%) |
| **AdaptInfer (Ours)** | 64 | FLOPs (T) | **0.975** | **0.959** | **1.317** | **1.305** | **1.139 ($\downarrow$ 74.3%)** |
| | | Latency (ms) | **31.0** | **30.9** | **34.5** | **35.5** | **33.0 ($\downarrow$ 61.3%)** |

Table 5: **Performance w/ and w/o dynamic text-guidance.** Comparisons are between static (SparseVLM) and dynamic text-guidance (Ours) methods under extreme pruning on LLava-1.5-7B.

| Method | Tokens | MME | TVQA | Ratio (%) |
|---|---|---|---|---|
| Vanilla | 576 | 1864 | 58.3 | 100 |
| SparseVLM | 48 | 1416 | 48.1 | 79.2 |
| **AdaptInfer** | 48 | **1494** | **52.8** | **85.4** |
| SparseVLM | 32 | 1098 | 47.7 | 70.4 |
| **AdaptInfer** | 32 | **1190** | **49.3** | **74.2** |

Table 6: **Performance of different pruning locations on LLava-1.5-7B with 128 tokens.** The chosen hyperparameters, inspired by our observations, yield the highest scores.

| Method | Pruning Loc. | MME | MMB | TVQA |
|---|---|---|---|---|
| **Ours** | **1,10,20** | **1794** | **63.8** | **56.8** |
| Uniform | 0,9,18,27 | 1788 | 62.8 | 56.4 |
| Uniform | 2,12,22 | 1758 | 63.2 | 56.5 |
| Single | 1 | 1668 | 60.8 | 56.3 |
| Random | - | 1679 | 62.1 | 55.8 |

## 4.5 ABLATION STUDY

Firstly, in order to illustrate the necessity of dynamic text guidance, we provide an experiment to compare results with a static text guidance work SparseVLM in Table 5. Also, we conduct this study as an exploration of extreme pruning. In the experiment, AdaptInfer consistently outperforms SparseVLM on MME and TextVQA (TVQA) under extreme low pruning budgets of only 48 and 32 vision tokens retained averagely. Specifically, AdaptInfer can keep a relatively high overall performance of 85.4% and 74.2% respectively. These results prove the effectiveness of our dynamic text guidance design.

Furthermore, we conduct an intuitive study to assess the effectiveness of our pruning hyperparameters. We compare our observation-driven pruning strategy with three baselines: uniform pruning, single-layer pruning, and random-layer pruning which averages results over five random configurations. The results shown in Table 6 confirm that the chosen hyperparameters are the best suited for AdaptInfer on the LLaMa-7B backbone.

## 5 CONCLUSION

This paper proposes a novel plug-and-play solution AdaptInfer for VLM acceleration via dynamic text-guided pruning. We further provide an offline analysis of cross-attention shifts, which motivates a principled pruning schedule. Our VLM acceleration plugin introduces minimal additional computational overhead while maintaining high accuracy. Specially, our AdaptInfer achieves SOTA accuracy under all the token budgets chosen in the experiments. For instance, AdaptInfer reduces CUDA latency by 61.3% and retains an average accuracy of 93.1% on LLaVA-1.5-7B, with only 64 vision tokens preserved per layer on average.

## REPRODUCIBILITY STATEMENT

We have submitted the source codes of our core algorithm in the supplementary materials to facilitate the reviewers' reproducibility check. Upon acceptance of this paper, we commit to releasing the full source code related to all experiments and results presented in the manuscript. In addition, we also provide more experimental details in the Appendix to improve the reproducibility.

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

## A  APPENDIX AND LLM USAGE STATEMENT

The chapters below are all technical appendix.

The authors declear that they use LLM tools for grammar refinement and language polishing only. The authors take full responsibility for the entire content of this manuscript.

## B  CHANGE-POINT DETECTION

To identify shifts in cross-attention across transformer layers, we employ change-point detection (Truong et al., 2020) on cumulative cross-attention curves. The intuition is that a significant change in attention behavior corresponds to a noticeable inflection in the cumulative attention distribution over layers.

Given a token's cross-attention curve across $L$ layers, denoted as:

$$\mathbf{a} = [a_1, a_2, \ldots, a_L], \quad \text{where } a_\ell \in \mathbb{R} \tag{8}$$

we first compute the cumulative attention:

$$\mathbf{c} = \left[ \sum_{i=1}^{1} a_i, \sum_{i=1}^{2} a_i, \ldots, \sum_{i=1}^{L} a_i \right] \tag{9}$$

This sequence $\mathbf{c}$ captures the aggregate build-up of attention, which we treat as a univariate time series. We then apply change-point detection to identify the most likely layer index where a shift in trend occurs.

We adopt the `ruptures` Python library[1] for offline detection. Specifically, we use the Binary Segmentation (Binseg) algorithm with an $\ell_2$ cost model. This algorithm aims to find a segmentation point that minimizes the within-segment variance. Formally, given the cumulative curve $\mathbf{c} = [y_1, y_2, \ldots, y_L]$, we solve:

$$\min_{b} \left( \sum_{t=1}^{b} (y_t - \mu_1)^2 + \sum_{t=b+1}^{L} (y_t - \mu_2)^2 \right), \tag{10}$$

where $\mu_1$ and $\mu_2$ are the mean values of the first and second segments, respectively. This corresponds to the $\ell_2$ cost model used in ruptures, which measures the homogeneity of each segment by its squared deviation from the mean.

We restrict the number of change points to 1, making the detection both efficient and robust. The returned breakpoint $b \in \{1, \ldots, L\}$ indicates the first significant shift in attention accumulation for the given token. This layer index is treated as the token's *cross-attention shift point*, which guides our downstream pruning strategy.

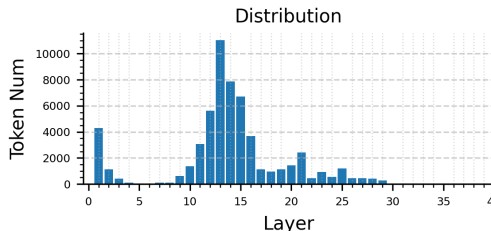

(a) Attention shift distribution on MME.

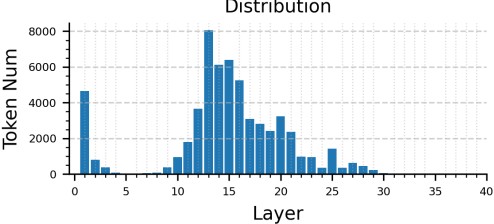

(b) Attention shift distribution on TextVQA.

Figure 4: **Layer-wise distribution of attention shifts on LLava-1.5-13B.**

| Tokens | MME | MMB | TextVQA | Ratio (%) |
|---|---|---|---|---|
| 576 | 1826 | 61.2 | 68.5 | 100 |
| 128 | 1813 | 59.8 | 67.4 | 98.5 |
| 64 | 1735 | 58.1 | 66.2 | 95.5 |

Table 7: **Performance of AdaptInfer on LLava-1.5-13B.** Our method still maintain high overall accuracy.

## C   EXPERIMENTS ON LLaVA-1.5-13B

### C.1   ATTENTION SHIFT DETECTION ON LLaVA-1.5-13B

To validate the transferability of our proposed pruning schedule, we conduct attention shift detection on LLaVA-1.5-13B (Liu et al., 2023b), which is built upon a 40-layer LLaMA-2-13B backbone (Touvron et al., 2023). The resulting distributions on two benchmark datasets, MME and TextVQA, are presented in Figure 4. We observe that the distribution of attention shift points still remains highly consistent on MME and TextVQA. The early-stage concentration of attention shifts consistently occurs at layer 1. Meanwhile, in later layers, dense change points are slightly delayed compared to those in LLaVA-1.5-7B, which typically occurs around layers 11 to 22. This suggests that for LLaVA-1.5-13B and other VLMs built on the LLaMA-2-13B backbone, an appropriate pruning schedule should select representative layers such as 1, 11, and 22.

Moreover, our attention shift detection pipeline is both simple and lightweight. For example, on the MME dataset, extracting 1000 attention maps and performing change-point detection takes only about 8 minutes in total. This confirms the efficiency and transferability of our pruning schedule across different model scales and architectures.

### C.2   PERFORMANCE OF ADAPTINFER ON LLava-1.5-13B

Using the parameters above, we test our AdaptInfer on LLava-1.5-13B to further validate the effectiveness of the proposed pruning schedule. The experiments are based on three widely used dataset MME (Fu et al., 2023), MMBench (MMB) (Liu et al., 2024b), and TextVQA (Singh et al., 2019) under 128 and 64 vision token budgets. The results are shown in Table 7.

Under the more demanding 13-billion-parameter setting of LLaVA-1.5, AdaptInfer continues to deliver high performance. As Table 7 shows, pruning the vision token numbers from 576 to 128 on average per layer results in only a 1.5% drop in overall accuracy (from 100% to 98.5%). Even with an aggressive 64 vision token budget, AdaptInfer achieves an 8.9× reduction in vision-token count, the model still retains 95.5% of its full-token accuracy. These numbers demonstrate that our token-adaptive pruning solution scales gracefully with model size, confirming AdaptInfer's robustness.

---

[1]https://github.com/deepcharles/ruptures

## D ATTENTION SHIFT DETECTION ON QWEN2-VL-2B

We also conduct our attention shift detection on a different VLM framework, Qwen2-VL-2B (Wang et al., 2024a), which contains 28 transformer layers in its LLM backbone. The attention shift distribution on MME is shown in Fig 5. The high-frequency regions are among layer 10-19, where stable regions are among layer 0-9 and 20-28. Based on our principled pruning schedule, we select the token pruning locations at layer 0, 9, and 19. Moreover, conducting the attention shift analysis on Qwen2-VL- 2B takes only about 5 minutes in total, which illustrates the lightweightness and transferability of the proposed attention shift analysis.

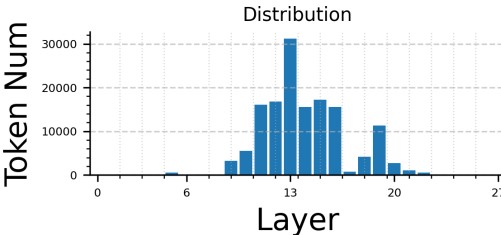

Figure 5: **Layer-wise distribution of attention shifts on Qwen2-VL-2B on MME.**

## E PRUNING TOLERANCE

To further understand the vision token pruning, we conduct a pruning-tolerance evaluation. Specifically, we apply AdaptInfer to Qwen2-VL-2B and report the performance across 14 multimodal tasks in the MME benchmark, comparing the scores with and without pruning. The results presented in Table 8 indicate that code reasoning, count and color-based tasks have a lower pruning tolerance relatively. Meanwhile, calculation and OCR tasks outperforms the original Qwen due to higher pruning tolerance and the noise reduction effect by token pruning.

Table 8: **Pruning-tolerance evaluation on MME.**

| Task | Qwen2-VL-2B | AdaptInfer (30% token retained) | Ratio |
|---|---|---|---|
| Existence | 200 | 200 | 100% |
| Count | 145 | 135 | 93.1% |
| Position | 158 | 158 | 100% |
| Color | 190 | 180 | 94.7% |
| Posters | 125 | 123 | 98.4% |
| Celebrity | 138 | 136 | 98.6% |
| Scene | 159 | 159 | 100% |
| Landmark | 174 | 172 | 98.9% |
| Artwork | 135 | 135 | 100% |
| OCR | 88 | 103 | 117% |
| Commonsense | 112 | 111 | 99.1% |
| Calculation | 23 | 33 | 143% |
| Translation | 163 | 155 | 95.1% |
| Code Reasoning | 93 | 85 | 91.4% |

## F INSTANCE-LEVEL DECISION ANALYSIS

There is an inherent trade-off between efficiency and performance in vision token pruning. we admit that aggressive token pruning can induce instance-level decision changes. To examine this effect, we perform a simple instance-level analysis comparing the vanilla Qwen2-VL-2B and 30% token retained AdaptInfer on MME. For each example, we record whether the prediction is (correct/error) before and after pruning. We tabulate the transition, and report the change ratios in Table 9. We find that the vast majority of predictions remain unchanged, while the fraction of Correct to Error cases is small and comparable to the number of Error to Correct cases.

Table 9: **Instance-Level change before and after pruning.**

|  | Correct to Correct | Correct to Error | Error to Correct | Error to Error |
|---|---|---|---|---|
| **Percentage** | 78.2% | 1.7% | 1.3% | 18.8% |

## G VISUALIZATION

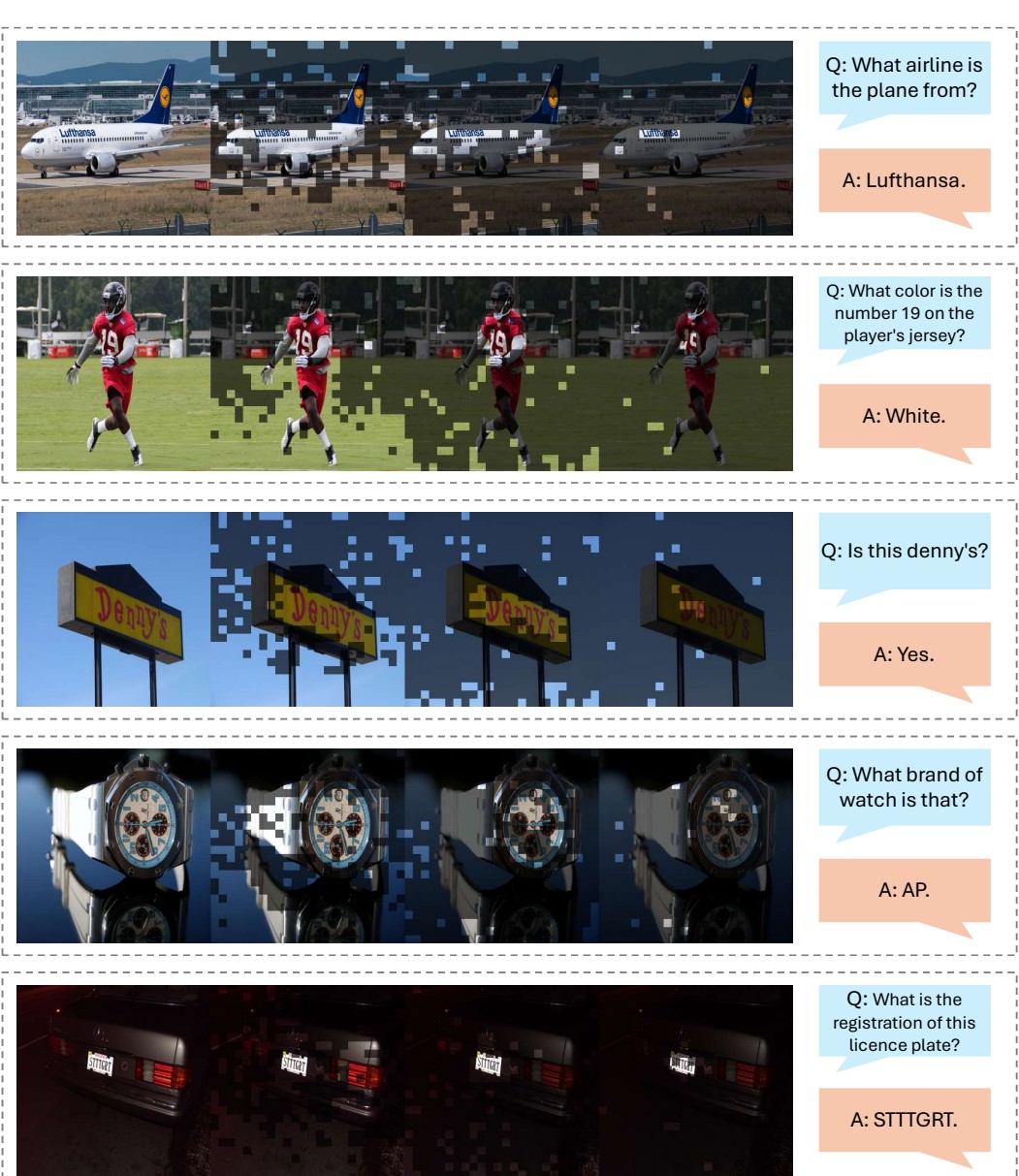

Figure 6: **Visualization of AdaptInfer on different examples.**

To qualitatively assess the pruning effect of the proposed AdaptInfer, we conduct an additional visualization study. In line with the main paper, vision tokens are pruned after the 1st, 10th, and 20th transformer layers. The resulting figures depict the spatial distribution of the remaining tokens at each stage: transparent regions correspond to tokens that survive pruning, whereas semi-transparent patches indicate positions that have been discarded.

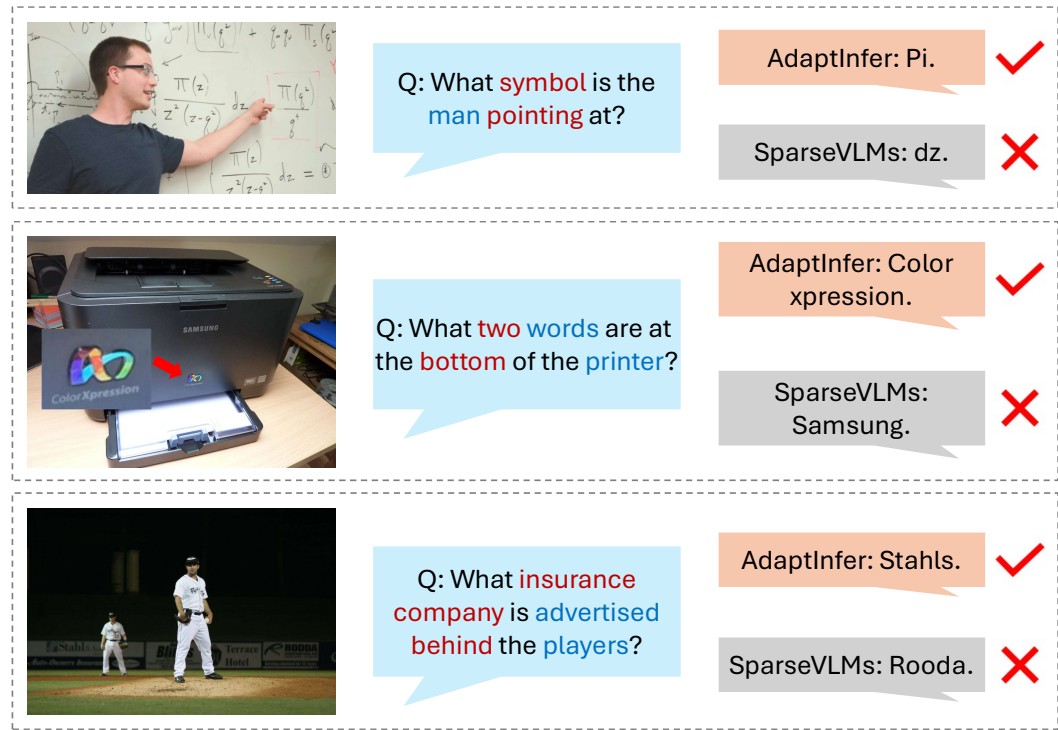

Figure 7: **Examples where AdaptInfer outperforms SparseVLM.** AdaptInfer performs better when text prompts are rich in descriptive modifiers.

The experiment is carried out on the TextVQA (Singh et al., 2019) benchmark, from which we sample a representative set of examples for display. As the layers deepen, ADAPTINFER rapidly suppresses tokens associated with semantically irrelevant background areas while consistently preserving those aligned with question-critical cues. The visualization is shown in Figure 6.

## H  PERFORMANCE ANALYSIS

We provide further analysis of the advantages of our dynamic text guidance method, AdaptInfer, compared with the static text guidance solution, SparseVLM (Zhang et al., 2025). We extract some VQA examples from the TextVQA dataset (Singh et al., 2019), and the answers of both methods are presented in Figure 7.

Within each question, blue highlights denote the static key text tokens selected offline by SparseVLM, whereas red highlights trace the key text tokens adaptively chosen from various layers by AdaptInfer. Close inspection reveals a consistent pattern: SparseVLM almost exclusively locks onto the text tokens that bear the strongest direct semantic link to the visual scene (e.g., *printer* or *players*), but often discards adjectival or adverbial modifiers that disambiguate entities (e.g., *pointing*, *two* or *behind*). When the prompt becomes syntactically richer and contains multiple content nouns and stacked modifiers static text guidance leads to an under-specified textual context and, consequently, a notable drop in answer accuracy. Conversely, AdaptInfer updates its key text token set dynamically during inference, rarely missing any prominent guidance. This dynamical adjustment make AdaptInfer overcome such limitation.

## I  EXPERIMENT SETUP

In this appendix section, we will briefly introduce the details of our experiment setups.

## I.1 Hardware Environment

To facilitate reproducibility, we ran every experiment on a single workstation that hosts an Intel Xeon Platinum 8358P processor together with eight NVIDIA RTX 4090 GPUs, each furnished with 24 GB of VRAM. The system operates under Ubuntu Linux 24.04, and all code was compiled using GCC 13.2.0 with Binutils 2.42 as the linker.

## I.2 Software Configuration

To improve reproducibility, we provide the main software configuration employed in our experiments. All runs were performed with the package versions listed below. We introduce the core deep-learning frameworks first, follow with performance-oriented extensions, and conclude with supporting libraries.

- **Python**: 3.10.18
- **PyTorch**: 2.1.2
- **Transformers**: 4.37.0
- **Accelerate**: 0.21.0
- **Flash-Attn**: 2.3.3
- **LLaVA**: 1.7.0.dev0
- **Tokenizers**: 0.15.1
- **TorchVision**: 0.16.2
- **ruptures**: 1.1.9

## I.3 General Implementation Details

In our evaluation of AdaptInfer, the pruning ratios at each pruning location are the hyperparameters need to be selected as well. For these hyperparameters, we follow the same rank-based sparsification-level adaptation method which SparseVLM (Zhang et al., 2025) introduced.

In the paper, we evaluated four vision–language models in total, including LLaVA-1.5-7B, LLaVA-1.5-13B, InternVL-Chat-7B (Chen et al., 2024a) and Qwen2-VL-2B. To stay within GPU-memory limits, we ran the LLava-7B variant and Qwen2-VL-2B on one GPU, the LLava-13B variant on three GPUs, and InternVL-Chat-7B on two GPUs (InternVL-Chat-7B contains a 6B-parameter vision encoder). All training and inference are performed in torch.float16 or torch.bfloat16 precision to further conserve memory usage. All the results of AdaptInfer reported in the context are the average of five runs.

## I.4 Implementation Details of AdaptInfer on Qwen2-VL-2B

Qwen2-VL-2B employs a dynamic number of vision tokens, ranging from 4 to 16,384 per image. To ensure stable performance and prevent GPU memory overflow, we constrain the vision token count within a bounded range, setting the minimum to 256 and the maximum to 1,280. For video-based multimodal evaluation, we follow FastV (Chen et al., 2025a) and SparseVLM (Zhang et al., 2025) and use the first 1,000 test samples from each benchmark to reduce evaluation time. The frame rate for all videos is fixed at 4 FPS.

## I.5 Compatibility with FlashAttention

During token ranking, we need t2t and t2v attentions to calculate the token importance scores. However, the original FlashAttention (Dao et al., 2022) does not allowed attention extraction. Fortunately, this issue has already been addressed by SparseVLM (Zhang et al., 2025), which provides a lightweight dual-pass solution that remains fully compatible with FlashAttention. Our method directly follows the same strategy.

Notably, because this solution can only get the mean t2t and t2v attention scores (averaged by multi-attention heads), we need to adjust our eqation 2 and 3 in the main paper into:

$$\mathbf{s} = \left(\sum_{i=1}^{T} \mathbf{A}_{t2t}^{(mean)}[i:]\right)^{\top} \cdot \mathbf{A}_{t2v}^{(mean)} \in \mathbb{R}^{V}. \tag{11}$$

### I.6 RANDOMNESS REPORT

Table 5 in the main paper contrasts our proposed *layer-wise pruning schedule* with three baselines: *uniform-layer pruning*, *single-layer pruning*, and *random-layer pruning*. For the random baseline, we executed five independent runs, each time drawing a different set of transformer layers to prune while keeping the token budget fixed. The sampled pruning locations are {3}, {2,15}, {2,8,16}, {2,4,8,16}, {3,6,23}.

