# OpenReview forum: "AdaptInfer: Adaptive Token Pruning for Vision-Language Model Inference with Dynamical Text Guidance"
_ICLR.cc/2026/Conference — Submitted to ICLR 2026_

### Official Review · Reviewer_7VKh · 2025-10-26

**Soundness:** 3
**Presentation:** 3
**Contribution:** 3
**Rating:** 4
**Confidence:** 3

**Summary:**

This paper proposes a training-free token pruning method. It directly leverages the attention computation mechanism of the base model for pruning. Furthermore, the paper provides insightful guidance on which layers are suitable for pruning and which are best left alone. The experiments in the paper are relatively comprehensive and provide strong support for the authors' claims.

**Strengths:**

The author uses experiments to heuristically verify which layers are more suitable for pruning and which are not. For example, the attention shift is more obvious in the 1st, 10th, and 20th layers.

**Weaknesses:**

1. The claims in 3.1.3 are based on experiments, but the authors tested them on only one model. They provide a discussion of generalizability, but there are limitations on whether these claims can be generalized to different models.
2. Tables 1 and 2 are too far removed from the analysis of the results to be easily readable.

**Questions:**

1. Based on the experiments in the author's methodology, I am curious whether this claim can be generalized to different tasks? For example, do classification tasks and QA tasks perform consistently across different layers? In addition, intuitively, classification tasks should be able to tolerate more aggressive pruning, so is it acceptable to prune layers where attention shift is obvious?
2. The method proposed by the author can prevent important information from being drop. Does this method ensure that the knowledge of the model remains consistent? As far as I know, token pruning methods can ensure that overall performance does not drop significantly. However, they may introduce knowledge boundary drift, meaning that answers that were previously correct may now be incorrect, and vice versa [1]. This possibility could have negative consequences in scenarios where critical questions must be answered correctly. If possible, could the authors conduct a simple analysis to see whether the proposed method leads to changes in individual case performance?

[1] Sun, Yizheng, et al. "Does Acceleration Cause Hidden Instability in Vision Language Models? Uncovering Instance-Level Divergence Through a Large-Scale Empirical Study." arXiv preprint arXiv:2503.06794 (2025).

---

> ### Author Response · Authors · 2025-11-26
> **Response to Reviewer 7VKh (1/2 parts)**
>
> ## **Response to Reviewer 7VKh**
>
> We sincerely thank you for the insightful and constructive comments. We provide our detailed responses to your comments below.
>
> ### **Weakness 1: Transferability of the Attention Shift Analysis and the Pruning schedule**
>
> Thank you for pointing out this. We conduct our attention-shift analysis of Qwen2-VL-2B on MME to further prove the transferability. We can still observe the distinct patterns on high-frequency and stable regions. The figure of the attention-shift distribution of Qwen2-VL is shown in Figure 5 in Appendix and we summarize it below:
>
> | Model             | High-frequency region | Stable regions         |
> |------------------|------------------------|-------------------------|
> | Qwen2-VL-2B      | 10–19                 | 0–9, 19–28
>
> Based on our proposed principle, we select pruning layers **0, 9, and 19**. We then evaluate AdaptInfer on Qwen2-VL-2B and the performance consistently matches or outperforms the latest SOTA baseline SparseVLM. The full results are shown in Table 3 of the revised manuscript.
>
> To further discuss, the necessity of performing this step stems from the distinct characteristics of each model. Besides, the proposed offline analysis is both transferable and computationally lightweight. Notably, the process requires only **eight minutes for LLaVA-1.5-7B and five minutes for Qwen2-VL-2B**. Although a new analysis is required for each model backbone, it greatly informs the selection of pruning parameters. Furthermore, this analysis is a one-time cost; once the model is deployed, it does not need to be repeated.
>
> ### **Weakness 2: Paper Layout**
> Thanks for your suggestion. We have adjusted the locations of the tables and made the paper layout to be easily readable in the revised manuscript.

---

> ### Author Response · Authors · 2025-11-26
> **Response to Reviewer 7VKh (2/2 parts)**
>
> ### **Question 1: Pruning Tolerance on Different Tasks**
>
> We agree with the reviewer that different tasks can exhibit different tolerance to pruning. Recent studies on visual token pruning in VLMs explicitly report task-dependent behavior [1]. For instance, ATP-LLaVA shows that a fixed pruning ratio can be either too aggressive or too conservative depending on the task difficulty and image content, and therefore advocates adaptive pruning strategies [2]. We also agree with the reviewer’s intuition that classification tasks may be able to tolerate more aggressive pruning, for the attentions on certain objects should be highly concentrated.
>
> Our evaluation already covers a broad range of multimodal tasks on different benchmarks, including **low-level visual perception tasks** (e.g., object recognition, OCR, counting) as well as **high-level reasoning tasks** (e.g., image understanding, scientific question answering, spatial reasoning). Among them, MME provide 14 different multimodal tasks, and specific evaluation scores on different task categories of Qwen2-VL-2B with AdaptInfer are shown below:
>
> | Task | Qwen2-VL-2B | AdaptInfer (30% token retained) | Ratio |
> |------------------|--------------------|----------------------|-------|
> | Existence | 200 | 200 | 100% |
> | Count | 145 |135 | 93.1% |
> | Position | 158 | 158 | 100% |
> | Color | 190 | 180 | 94.7% |
> | Posters | 125 | 123 | 98.4% |
> | Celebrity | 138 | 136| 98.6% |
> | Scene | 159 | 159| 100% |
> | Landmark | 174 | 172| 98.9%|
> | Artwork | 135 | 135 | 100% |
> | OCR | 88 | 103 | 117% |
> | Commonsense | 112 | 111 | 99.1% |
> | Calculation | 23 | 33 | 143% |
> | Translation | 163 | 155 | 95.1% |
> | Code Reasoning| 93 | 85 | 91.4% |
>
> The results above indicate that code reasoning, count and color-based tasks have a lower pruning tolerance relatively. Meanwhile, calculation and OCR tasks outperform the original Qwen due to higher pruning tolerance and even benifit from the noise reduction effect by token pruning.
>
> [1] Z. Wen et al., “Token Pruning in Multimodal Large Language Models: Are We Solving the Right Problem?” in Findings of ACL, pp. 15537–15549, Vienna, 2025.
>
> [2] X. Ye et al., “ATP-LLaVA: Adaptive Token Pruning for Large Vision Language Models,” in Proc. CVPR, pp. 24972–24982, 2025.
>
> ### **Question 2: Knowledge Boundary Drift and Instance-Level Analysis**
>
> We appreciate the reviewer for raising this important point. Our method is training-free and does not modify any model parameters or architectures. It only drops unimportant vision tokens during inference. Thus, it does not change the knowledge of the model. However, we agree that aggressive token pruning can still induce instance-level decision changes (i.e., knowledge boundary drift), as also observed in [1]. This is an inherent trade-off between efficiency and performance.
>
> To examine this effect, we perform a simple instance-level analysis comparing the vanilla Qwen2-VL-2B and 30% token retained AdaptInfer on MME. For each example, we record whether the prediction is (correct/error) before and after pruning. We tabulate the transition in Table 9 in Appendix, and report the change ratios below:
>
> | Correct to Correct| Correct to Error| Error to Correct | Error to Error |
> |------------------|------------------|------------------|------------------|
> | 78.2% | 1.7% | 1.3% | 18.8% |
>
> We find that the vast majority of predictions remain unchanged, while the fraction of Correct to Error cases is small and comparable to the number of Error to Correct cases. These results suggest that, under the pruning regimes we mainly recommend, AdaptInfer preserves model behavior on most individual instances while occasionally correcting or degrading borderline cases.
>
> [1] Y. Sun et al., “Does Acceleration Cause Hidden Instability in Vision Language Models? Uncovering Instance-Level Divergence Through a Large-Scale Empirical Study,” arXiv preprint arXiv:2503.06794, 2025.

---

> ### Comment · Reviewer_7VKh · 2025-11-28
>
> Thank you for the authors’ timely and thorough response, as well as the substantial additional experiments. I have reviewed the revised papers, and I hope that these improvements can be incorporated into the next version of the paper.

---

> > ### Author Response · Authors · 2025-11-28
> >
> > Thank you very much for your recognition. Your suggestions have greatly improved the quality of our work, and all these improvements will be incorporated in the new version of the paper. If there are any remaining concerns or additional suggestions, we would be very happy to further address them. We hope that our responses have satisfactorily resolved all your questions.

---

### Official Review · Reviewer_nCEm · 2025-10-27

**Soundness:** 2
**Presentation:** 2
**Contribution:** 1
**Rating:** 2
**Confidence:** 4

**Summary:**

This paper presents AdapterInfer, a plug-and-play training-free adaptive token pruning approach for VLMs. First, they use layer-wise text-to-text attention map to enhance the text-based vision-token pruning, use important text tokens to mine important vision tokens. Second, they identify consistent inflexion locations in inference via analysis of cross-modal attention in LLaVA, and improve the pruning schedule.

**Strengths:**

1. **Reasonable Design.** Using important text tokens to mine important vision tokens is reasonable.

**Weaknesses:**

1. **Weak Baseline.** This paper uses LLaVA-1.5 as a baseline, which is an open-source VLM released 2 years ago. It's too weak and completely outdated. For a training-free method, the evaluation should be conducted on more recent, competitive and widely used VLMs, such as Qwen2.5-VL.

2. **Lack Generalizability.** The consistent attention inflection point found in this paper is based on LLaVA-1.5. The author should check other base VLMs like Qwen2-VL and Qwen2.5-VL for a similar situation.

3. **Lack Comparison.** This paper should compare with newer baselines, like VisionZip[1], a widely used efficient VLM approach accepted in CVPR25.

4. **Cannot work with flash-attention.** AdapterInfer based on t2t and t2v attention score, which cannot be obtained when inference with flash-attention.

[1] VisionZip: Longer is Better but Not Necessary in Vision Language Models

**Questions:**

1. **Reasoning VLMs.** Reasoning VLMs also face the problem of token redundancy. The author should discuss whether this method can be applied to Efficient Reasoning VLMs[1].

2. **Resize.** In [1], the researchers discussed that in most general scenarios, even simple resizing can achieve strong performance. How do the authors view this issue? I look forward to the author discussing this point.

[1] VisionThink: Smart and Efficient Vision Language Model via Reinforcement Learning

---

> ### Author Response · Authors · 2025-11-26
> **Response to Reviewer nCEm (1/2 parts)**
>
> ##**Response to Reviewer nCEm**
>
> We sincerely thank you for the insightful and constructive comments. We provide our detailed responses to your comments below.
>
> ### **Weakness 1 & 2: Experiments on Newer VLM**
>
> Thanks for your suggestion. We agree that evaluating AdaptInfer on newer VLMs can further strengthen the empirical evidence. In the revised manuscript, we therefore include **additional experiments on Qwen2-VL-2B**, a more recent and robust open-source VLM that supports both image-based and video-based QA tasks.
>
> We conduct our lightweight, transferable attention-shift analysis of Qwen2-VL-2B on MME, the attention shift distribution is shown in Figure 5 in Appendix, and we summarize it below:
>
> | Model             | High-frequency region | Stable regions  |
> | - | - | - |
> | Qwen2-VL-2B      | 10–19                 | 0–9, 19–28   |
>
> Based on our proposed principle, we select pruning layers **0, 9, and 19**. We then evaluate AdaptInfer on **8 benchmarks**, including **3 video QA datasets** (TGIF-QA, MSRVTT-QA, MSVD-QA), providing a broader and more challenging comparison.
>
> Unlike LLaVA, Qwen2-VL dynamically adjusts its vision token counts. Thus, for fairness, we compare SparseVLM and AdaptInfer under the **same retained-token ratios** (50%, 30%, 10%). The results are summarized below:
>
> | Methods      | Tokens | MME | TextVQA | GQA | MMB | POPE | TGIF-QA | MSRVTT-QA | MSVD-QA | Overall Ratio |
> |--------------|--------|-----|---------|-----|-----|------|------|--------|------|----------------|
> | Ori-Qwen     | 100%   | 1901 | 77.8 | 60.4 | 71.7 | 86.9 | 9.9 | 30.9 | 42.3 | 100% |
> | SparseVLM    | 50%    | 1900 | 76.9 | 59.9 | 71.0 | 86.4 | 11.7 | 31.2 | 42.9| 102.1% |
> | AdaptInfer   | 50%    | 1909 | 76.6 | 60.0 | 72.1 | 86.7 | 11.1 | 30.5 | 43.6 | 101.7% |
> | SparseVLM    | 30%    | 1867 | 73.6 | 57.5 | 70.1 | 84.6 | 9.3 | 29.5 | 41.8 | 96.4% |
> | AdaptInfer   | 30%    | 1885 | 73.9 | 58.5 | 71.4 | 85.8 | 10.4 | 30.2 | 43.5 | 99.4% |
> | SparseVLM    | 10%    | 1460 | 51.3 | 40.4 | 57.3 | 40.3 | 5.0 | 28.1 | 30.2 | 68.6% |
> | AdaptInfer   | 10%    | 1721 | 62.3 | 48.8 | 63.7 | 71.3 | 5.9 | 28.2 | 41.4 | 81.6% |
>
> Across both image-based QA and **multi-frame video QA**, AdaptInfer consistently matches or outperforms SparseVLM. Our method performs better particularly under the **30%** and **10%** retained-token settings, where the pruning becomes more aggressive.
>
> Interestingly, at the **50% retained ratio**, both SparseVLM and AdaptInfer slightly outperform the original model. We believe this reflects a widely observed property in multimodal QA: the presence of a large number of redundant visual tokens. Removing these distractors not only accelerates inference but can also *improve* QA accuracy by reducing noise. This further confirms the strong practical value of token pruning.
>
> Moreover, these results highlight an important distinction in token importance ranking. Under mild pruning (e.g., 50%), all methods remove the most obviously redundant tokens, leading to similar performance. However, under aggressive pruning (30% and 10%), the ranking quality becomes critical so that the advantages of our **dynamic text-guided ranking** emerge clearly, yielding significantly stronger performance.
>
> Overall, this extended evaluation demonstrates that AdaptInfer generalizes well across architectures, modalities, and task types, providing a more comprehensive and convincing comparison.
>
> ### **Weakness 3: Baseline VisionZip**
>
> Thank you for pointing this out. As discussed in Section 4.1.2, **VisionZip operates on a fundamentally different stage of the VLM pipeline**: it prunes and compresses tokens *inside the vision encoder*, whereas our method, along with PyramidDrop and SparseVLM, focuses on **pruning during the LLM prefill stage**. This makes VisionZip *orthogonal* to our approach rather than a direct baseline. In fact, VisionZip can be **combined** with AdaptInfer to produce cumulative benefits.
>
> We appreciate the insights introduced by VisionZip: it aims to distill redundant information within visual embeddings, producing a compact set of informative tokens before entering the language model. However, this encoder-side compression is **prompt-agnostic** —the visual tokens are compressed without considering the text query.
>
> In contrast, context-aware pruning approaches such as **PyramidDrop, SparseVLM, and ours**explicitly model how *token importance depends on the prompt*, aligning more closely with how humans perform multimodal reasoning: we read the question first, then selectively attend to the most relevant regions in the image. Different prompts can correspond to very different visual importance patterns, which encoder-only pruning cannot adapt to.
>
> For these reasons, we view VisionZip not as a baseline for direct comparison, but as a **complementary technique** that could potentially be integrated with our method in future work.

---

> ### Author Response · Authors · 2025-11-26
> **Response to Reviewer nCEm (2/2 parts)**
>
> ### **Weakness 4: FlashAttention**
>
> We thank the reviewer for raising this concern. Fortunately, this issue has already been addressed by **SparseVLM**, which provides a lightweight solution that remains fully compatible with FlashAttention. Our method directly follows the same strategy. The key idea is to introduce a *dual-pass FlashAttention*:
>
> 1. **First pass:** run standard FlashAttention to obtain hidden states as usual.
> 2. **Second pass:** inject a specially constructed **V matrix** where the rows corresponding to text token indexes are set to a constant mean weight.
>
> This allows FlashAttention to directly output the **mean attention scores** from text tokens to each themselves and visual token without materializing the attention matrix.
>
> This mechanism is efficient, requires no modification to FlashAttention itself, and does not store or compute any large attention matrix. Since AdaptInfer can also only depend on mean t2t/t2v attention scores, not on the raw matrix, it can be implemented using exactly the same dual-pass trick. Thus, AdaptInfer **is compatible with FlashAttention**, just as SparseVLM has demonstrated.
>
> ### **Question 1: Token Pruning in Reasoning VLMs**
>
> To the best of our knowledge, reasoning fall into two main categories in VLMs. The first category focuses on **language-only reasoning** [1] where the visual input is used only once at the beginning and the subsequent reasoning are purely using texts. Our method is not directly applicable to these models because AdaptInfer specifically targets **vision-token redundancy** during the LLM prefilling.
>
> The second category involves **vision-informed reasoning**, where the model repeatedly generates intermediate visual states as part of the chain-of-thoughts. Then, those vision tokens can be generated into images via diffusion models [2]. These approaches naturally produce a large number of redundant visual tokens, making them well aligned with our pruning objective. AdaptInfer can be readily applied here to reduce visual-token overhead at each reasoning step, improving efficiency without altering the model’s reasoning mechanism.
>
> We also note an emerging line of work where reasoning is performed in the **latent visual-space** instead of generating explicit image [3]. Such models still accumulate substantial redundant visual representations through iterative reasoning, and therefore could benefit even more from periodic vision-token pruning to prevent excessively long token sequences and reduce computation.
>
> Overall, AdaptInfer is particularly suitable for vision-informed reasoning pipelines, where visual tokens are accumulated over time.
>
> [1] Z. Mi et al., “I Think, Therefore I Diffuse: Enabling Multimodal In-Context Reasoning in Diffusion Models,” in Proc. ICML, 2025.
>
> [2] Z. Zhang et al., “Multimodal Chain-of-Thought Reasoning in Language Models,” Trans. Mach. Learn. Res., 2024.
>
> [3] Z. Yang et al., “Machine Mental Imagery: Empower Multimodal Reasoning with Latent Visual Tokens,” arXiv preprint arXiv:2506.17218, 2025.
>
> ### **Question 2: Image Resizing**
>
> Thank you for the comment. We appreciate the observation from VisionThink that simple resizing can already improve efficiency in some scenarios. This highlights that many multimodal tasks indeed contain substantial visual redundancy.
>
> However, resizing is inherently **task-agnostic**: it reduces both useful details and irrelevant visual noise uniformly. In contrast, approaches like VisionZip or ours aim to remove **low-information or prompt-irrelevant tokens**, preserving key content while discarding noise.
>
> This selectivity is important for multimodal QA, where the relevance of visual regions depends on the text query. As a result, context-aware pruning can be more efficient and **sometimes even improve accuracy**. For example, AdaptInfer reaches **101.7% average performance** under a token pruning ratio of 50% on Qwen2-VL-2B.
>
> Overall, we view resizing as a simple and effective baseline, while our method provides a complementary, more targeted way to remove redundancy.

---

### Official Review · Reviewer_KLT6 · 2025-11-01

**Soundness:** 3
**Presentation:** 3
**Contribution:** 2
**Rating:** 4
**Confidence:** 3

**Summary:**

This paper proposes an adaptive token pruning method for VLMs, of which the core design is a plug-and-play framework to reuse the text-to-text attention maps for a soft prior of text-token importance. In the experiments, The proposed method outperforms FastV, SparseVLM, and other baselines on the average accuracy of 5 VQA benchmarks, showing the effectiveness of the AdaptInfer module.

**Strengths:**

- The proposed dynamic cross-attention guided visual token pruning is enhanced by reusing text-token attantion to put a higher focus on important text tokens.
- The proposed AdaptInfer leverage the observation of attention distribution shift to guide choices of hyperparameters like the insertion location of the pruning layer.
- The proposed method provides notable savings in inference latency.

**Weaknesses:**

- The proposed method is only verified on 5 different VQA datasets, whereas the baseline methods are usually evaluated on much more diverse benchmarks. For example the PyramidDrop is evaluated on 16 different benchmarks.
- Some benchmarks are also evaluated on video VQA benchmarks, including PyramidDrop and SparseVLM. A more well-rounded comparison will be more convincing especially when the improvement over SparseVLM is not consistent.
- The newer SOTA benchmark VisionZip [1] is not included in the comparison.
[1] Yang, Senqiao, Yukang Chen, Zhuotao Tian, et al. n.d. VisionZip: Longer Is Better but Not Necessary in Vision Language Models.

**Questions:**

See weaknesses.

---

> ### Author Response · Authors · 2025-11-26
> **Response to Reviewer KLT6**
>
> ## **Response to Reviewer KLT6**
>
> We sincerely thank you for the insightful and constructive comments. We provide our detailed responses to your comments below.
>
> ### **Weakness 1 & 2: Additional Benchmarks**
>
> Thank you for the valuable suggestion. We agree that evaluating on more diverse benchmarks can further strengthen the empirical evidence. In the revised version, we therefore include **additional experiments on Qwen2-VL-2B**, a recent and robust open-source VLM that supports both image-based and video-based QA tasks.
>
> Based on our attention-shift analysis of Qwen2-VL-2B, we select pruning layers **0, 9, and 19**. The distribution is shown in Figure 5 in Appendix. We then evaluate AdaptInfer on 8 benchmarks, including **3 video QA datasets** (TGIF-QA, MSRVTT-QA, MSVD-QA), providing a broader and more challenging comparison.
>
> Unlike LLaVA, Qwen2-VL dynamically adjusts its vision token counts. Thus, for fairness, we compare SparseVLM and AdaptInfer under the **same retained-token ratios** (50%, 30%, 10%). The results are shown in the Table 3 of the revised manuscript and are summarized below:
>
> | Methods      | Tokens | MME | TextVQA | GQA | MMB | POPE | TGIF-QA | MSRVTT-QA | MSVD-QA | Overall Ratio |
> |--------------|--------|-----|---------|-----|-----|------|------|--------|------|----------------|
> | Ori-Qwen     | 100%   | 1901 | 77.8 | 60.4 | 71.7 | 86.9 | 9.9 | 30.9 | 42.3 | 100% |
> | SparseVLM    | 50%    | 1900 | 76.9 | 59.9 | 71.0 | 86.4 | 11.7 | 31.2 | 42.9| 102.1% |
> | AdaptInfer   | 50%    | 1909 | 76.6 | 60.0 | 72.1 | 86.7 | 11.1 | 30.5 | 43.6 | 101.7% |
> | SparseVLM    | 30%    | 1867 | 73.6 | 57.5 | 70.1 | 84.6 | 9.3 | 29.5 | 41.8 | 96.4% |
> | AdaptInfer   | 30%    | 1885 | 73.9 | 58.5 | 71.4 | 85.8 | 10.4 | 30.2 | 43.5 | 99.4% |
> | SparseVLM    | 10%    | 1460 | 51.3 | 40.4 | 57.3 | 40.3 | 5.0 | 28.1 | 30.2 | 68.6% |
> | AdaptInfer   | 10%    | 1721 | 62.3 | 48.8 | 63.7 | 71.3 | 5.9 | 28.2 | 41.4 | 81.6% |
>
> Across both image-based QA and **multi-frame video QA**, AdaptInfer consistently matches or outperforms SparseVLM. Our method performs better particularly under the **30%** and **10%** retained-token settings, where the pruning becomes more aggressive.
>
> Interestingly, at the **50% retained ratio**, both SparseVLM and AdaptInfer slightly outperform the original model. We believe this reflects a widely observed property in multimodal QA: the presence of a large number of redundant visual tokens. Removing these distractors not only accelerates inference but can also *improve* QA accuracy by reducing noise. This further confirms the strong practical value of token pruning.
>
> Moreover, these results highlight an important distinction in token importance ranking. Under mild pruning (e.g., 50%), all methods remove the most obviously redundant tokens, leading to similar performance. However, under aggressive pruning (30% and 10%), the ranking quality becomes critical so that the advantages of our **dynamic text-guided ranking** emerge clearly, yielding significantly stronger performance.
>
> Overall, this extended evaluation demonstrates that AdaptInfer generalizes well across architectures, modalities, and task types, providing a more comprehensive and convincing comparison.
>
> ### **Weakness 3: Baseline VisionZip**:
> Thank you for pointing this out. As discussed in Section 4.1.2, **VisionZip operates on a fundamentally different stage of the VLM pipeline**: it prunes and compresses tokens *inside the vision encoder*, whereas our method, along with PyramidDrop and SparseVLM, focuses on **pruning during the LLM prefill stage**. This makes VisionZip *orthogonal* to our approach rather than a direct baseline. In fact, VisionZip can be **combined** with AdaptInfer to produce cumulative benefits.
>
> We appreciate the insights introduced by VisionZip: it aims to distill redundant information within visual embeddings, producing a compact set of informative tokens before entering the language model. However, this encoder-side compression is **prompt-agnostic** —the visual tokens are compressed without considering the text query.
>
> In contrast, context-aware pruning approaches such as **PyramidDrop, SparseVLM, and ours**explicitly model how *token importance depends on the prompt*, aligning more closely with how humans perform multimodal reasoning: we read the question first, then selectively attend to the most relevant regions in the image. Different prompts can correspond to very different visual importance patterns, which encoder-only pruning cannot adapt to.
>
> For these reasons, we view VisionZip not as a baseline for direct comparison, but as a **complementary technique** that could potentially be integrated with our method in future work.

---

### Official Review · Reviewer_mo21 · 2025-11-01

**Soundness:** 3
**Presentation:** 3
**Contribution:** 3
**Rating:** 4
**Confidence:** 3

**Summary:**

The authors propose "AdaptInfer," a plug-and-play, training-free framework designed to address these two specific problems.

First, the core technical novelty is a dynamic text-guided pruning mechanism. Instead of using a static set of "key" text tokens, AdaptInfer re-computes text-token importance at each designated pruning layer. It cleverly reuses the existing text-to-text (t2t) attention maps (which are already computed) to create a "soft prior" distribution over the text tokens. This prior is then used to reweight the text-to-vision (t2v) attention, yielding a more informed and context-aware ranking of vision tokens for pruning. This is an elegant solution, as it introduces minimal computational overhead.

Second, to replace heuristic pruning schedules, the authors conduct an offline analysis to identify "attention inflection points." By applying change-point detection to the cumulative t2v attention trajectories, they discover consistent, data-driven locations in the architecture (e.g., layers 1, 10, and 20 for LLaVA-1.5-7B) where the model's utilization of visual information significantly shifts. These inflection points are then used as principled locations for applying the pruning.

**Strengths:**

- The method is both training-free and, critically, reuses attention maps that are already computed during the forward pass. This means it introduces almost no additional overhead (Sec 3.2.2), making it an extremely practical solution for real-world inference acceleration.

- The paper shows clear and consistent performance gains over its closest SOTA-level competitor (SparseVLM) across multiple benchmarks and token budgets (Table 1, Fig 3). The latency test (Table 3) confirms that these theoretical gains translate to real-world speedups.

- The primary contribution—using dynamic text-to-text attention as a proxy for text-token importance—is a novel and elegant solution to the limitations of static guidance. It's theoretically well-motivated by the (correct) observation that information importance evolves by layer.

**Weaknesses:**

- The specific schedule (layers 1, 10, 20) is architecture-dependent (LLaMA-7B). While the method for finding the schedule is general, it requires a new, non-trivial offline analysis (running 1000+ samples through the model and performing change-point detection) for every new backbone

- The main paper emphasizes the locations of pruning, but the amount to prune at each location is also a critical hyperparameter. Appendix G.3 mentions that these "pruning ratios" are also selected, following a method from prior work. The sensitivity of the model to these ratios, and how they interact with the pruning locations, is not fully explored, adding a layer of tuning that is under-discussed.

- In Sec 3.1.3, the authors posit that an attention shift can mean a token is either becoming critical or redundant. The logic in Sec 3.2.3 for building the schedule around these points ("pruning should avoid the regions with intense attention shifts... pruning just before or after the dense regions") feels slightly post-hoc and could be articulated more clearly. For instance, why is pruning at layer 1 (an inflection point) and at layer 10 (the start of a dense region) the optimal strategy?

**Questions:**

The questions are mainly covered by the weakness above.

---

> ### Author Response · Authors · 2025-11-26
> **Response to Reviewer mo21**
>
> ## **Response to Reviewer mo21**
> We sincerely thank you for the insightful and constructive comments. We provide our detailed responses to your comments below.
>
> ### **Weakness 1: Attention Shift Analysis**
>
> The necessity of performing this step stems from the distinct characteristics of each model. Besides, the proposed offline analysis is both transferable and computationally lightweight. Notably, the process requires only eight minutes for LLaVA-1.5-7B and five minutes for Qwen2-VL-2B. Although a new analysis is required for each model backbone, it greatly informs the selection of pruning parameters. Furthermore, this analysis is a one-time cost; once the model is deployed, it does not need to be repeated. This approach is far more efficient than the labor-intensive practice of manually tuning on evaluation sets to find optimal settings, as seen in many prior works.
>
> ### **Weakness 2: Pruning Ratios**
>
> Thank you for the insightful comment. We agree that pruning ratios constitute an important hyperparameter in addition to pruning locations. In our work, we follow the pruning-ratio selection strategy introduced in **SparseVLM (Zhang et al., 2025)**, as our contribution focuses on *where* to prune rather than *how much* to prune.
> SparseVLM determines the pruning ratios based on the **rank of the layer-wise text-to-vision (t2v) attention matrix**, under the assumption that a full-rank matrix indicates reduced redundancy. Specifically, the retained number of visual tokens at each pruning layer is:
>
> $$
> N = \lambda \times (N_v - \text{rank}(A_{t2v}))
> $$
>
> where \(N_v\) is the number of visual tokens before pruning, and \(\lambda\) is a global scaling coefficient. In our implementation, **λ is kept fixed across all pruning layers**. Once the target *average retained token budget* (e.g., 128, 64) is specified, λ becomes uniquely determined. While we agree that analyzing sensitivity to these ratios may provide additional insights, it is orthogonal to our proposed contributions. Since our method does not modify the ratio-selection policy itself, keeping the ratio mechanism unchanged ensures that any performance gains arise solely from our contributions rather than additional hyperparameter tuning.
>
> ### **Weakness 3: Pruning Location Selection**
>
> Pruning visual tokens introduces an inherent trade-off between accuracy and efficiency. To prune aggressively, one must prune early; yet pruning too early inevitably risks removing tokens whose importance has not yet emerged.
> Prior works rely directly on attention scores to rank tokens. However, we argue that attention scores from all transformer layers are not equally reliable for serving as evidence for pruning. Therefore, we provide an analysis on **attention-shift distribution**, which can indicate how reliable the attention-based rankings are across layers.
>
> Our attention-shift analysis reveals two schemes:
> - **High-frequency regions** (e.g., layer 1 and layers 10–20 on LLaVA-1.5-7B), where token importance is actively being reassigned (either becoming important for the first time or being used up by the model); attention rankings here are unstable and thus unsafe for pruning.
> - **Stable regions** (e.g., layers 2–9 and 20+), where importance rankings remain consistent and therefore provide reliable pruning signals.
>
> Based on this, pruning **after layer 1** and **after layer 20** on Llava-1.5-7B is well justified. Both locations mark the beginning of each stable region, enabling early pruning to save more FLOPs relying on stable attention rankings.
>
> Pruning after layer 10 does not rely on the attention-shift patterns. Instead, it serves as a necessary compromise to prevent over-pruning at layer 1, which would remove too many informative tokens too early. For this reason, we choose an additional pruning location between layers 1 and 20. Placing this step roughly midway (layer 10) ensures both sufficient depth for each pruning to take effect and little interference with the high-volatility band.
>
> In summary, pruning after layers 1 and 20 is directly supported by our attention-shift analysis, while pruning at layer 10 is a principled compromise for balancing safety and efficiency. Importantly, our approach is designed to avoid dataset-specific tuning for selecting pruning locations, unlike many prior works that rely heavily on empirical search.
>
> To confirm that the above principle generalizes beyond LLaMA-based VLMs, we conducted the same attention-shift analysis on **Qwen2-VL-2B**. The distribution again separates layers into distinct regimes:
>
> | Model             | High-frequency region | Stable regions         |
> | - | - | - |
> | Qwen2-VL-2B      | 10–19                 | 0–9, 19–28            |
>
> Following the same principle, we choose pruning layers **0, 9, and 19** for Qwen2-VL-2B. The distribution is shown in Figure 5 in Appendix and the resulting pruning-inference performance is reported in the Table 3 of the revised manuscript .

---

### Author Response · Authors · 2025-11-27
**Revision**

We sincerely thank all reviewers, as well as the ACs, SACs, and PCs, for your time and effort evaluating our work and the valuable comments provided. Based on the review, we have revised our manuscript to further strengthen the clarity and quality. Our main modifications are as follows:

1.	Advised by reviewer KLT6 and nCEm, we provided further experiments on a newer VLM Qwen2-VL and on more benchmarks, including video-based benchmarks (Section 4.3).

2.	Advised by reviewer mo21, cCEm and 7VKh, we conducted additional attention shift analysis on Qwen2-VL, showing the transferability of the proposed analysis and pruning schedule (Section 3.3 and Appendix D).

3.	Advised by reviewer 7VKh, we added new experiments on exploring pruning tolerance and instance-level decision changes, to provide a deeper understanding of vision token pruning.

4.	Advised by reviewer nCEm, we briefly discussed the compatibility with FlashAttention (Appendix I.5).

5.	Advised by reviwer 7VKh, we modified the locations of some tables and the paper layout to make it easier to read.

We use blue text to highlight the modification. We hope these revisions and our specific responses address the reviewers’ concerns. We also look forward to any deeper discussion, which will undoubtedly help us continue improving the quality of our work.

---

### Author Response · Authors · 2025-12-03
**Global Summary**

We appreciate all the ACs, SACs and PCs for the time and effort reviewing our work. To facilitate your assessment, we provide a summary of the main concerns of each reviewer and how we have addressed each of the concerns.

**For Reviewer mo21,** the main concerns include (1) the architecture-specific attention-shift analysis requirement, (2) the method for determining pruning ratios, and (3) the clarity of the reasons for the selected pruning locations.

***Our Response:***

(1) We clarified that the offline attention-shift analysis is a necessary and low-cost step that provides valuable guidance for constructing pruning schedules.

(2) We detailed the pruning-ratio determination process, which is adopted from prior work, noting that our method utilizes—rather than modifies—this component.

(3) We provided a more thorough explanation of the pruning-location selecting principles, supported by attention-shift patterns and updated in the revised manuscript.

**For Reviewer KLT6,** the main concerns focus on (1) the breadth of benchmark coverage, and (2) Lack of comparison with baseline VisionZip.

***Our Response:***

(1) We expanded our evaluation by incorporating the more recent and widely adopted Qwen2-VL, and additionally reported results on broader benchmarks, including three video-based datasets.

(2) We clarified that VisionZip is orthogonal to our proposed approach, and its relationship to our method had been already discussed in Section 4.1.2 of the manuscript.

**For Reviewer nCEm,** the main concerns and questions are about (1) the breadth of VLM baseline coverage, (2) the transferability of the attention shift detection, (3) Lack of comparison with baseline VisionZip, (4) potential incompatibility with FlashAttention, (5) token pruning in reasoning VLMs and (6) the effect of resizing on reducing token counts.

***Our Response:***

(1) We strengthened the evaluation by incorporating the more recent and widely adopted VLM Qwen2-VL and additionally reported results on broader benchmarks.

(2) We provided attention shift analysis on Qwen2-VL, demonstrating the transferability of the method.

(3) We clarified that VisionZip is orthogonal to our proposed approach, and its relationship to our method had been already discussed in Section 4.1.2 of the manuscript.

(4) We described existing solutions that enable compatibility with FlashAttention.

(5) We discussed the deployment considerations for different types of reasoning VLMs.

(6) We also shared our insights on this topic.

**For Reviewer 7VKh,** the main concerns and questions are about (1) the transferability of the attention shift detection, (2) the manuscript layout, (3) the pruning tolerance across different tasks, and (4) potential knowledge-boundary drift and instance-level behavior changes.

***Our Response:***

(1) We conducted attention shift analysis on Qwen2-VL, demonstrating the transferability of the method.

(2) We revised and improved the manuscript layout accordingly.

(3) We discussed the pruning tolerance across task types and provided additional experiments to support the observations.

(4) We offered our insights and included an analysis of instance-level changes in answers.

---

> ### Author Response · Authors · 2025-12-03
>
> In summary, we provided detailed, point-by-point responses to all reviewer concerns. We believe that we have adequately addressed most of the issues raised by the reviewers, as reflected in the positive feedback from Reviewer 7VKh.
>
> We hope this summary and the revisions are helpful in your decision-making process. Thank you once again for your effort and support. Best wishes.

---

### Meta-Review · Area_Chair_JjLs · 2025-12-20

**Summary:**

This paper received 4,4,2,4.  The main concerns include questions regarding the generality of the proposed approach across different architectures, some heuristic and unclear approach design choices, and missing comparisons to more recent SOTA baselines and needing more experiments to adequately validate the approach.  These were partially addressed by the rebuttal (details in next section).

**Reviewer Concerns:**

Overall, the concerns regarding heuristic and unclear approach design choices are largely addressed by the rebuttal, through clearer explanations. The remaining concerns around generalization to different architectures and experimental validation were partially addressed, but not fully. The rebuttal correctly notes that that VisionZip is orthogonal to the proposed approach, although the claim that it could be complementary would be more compelling with an experimental validation. The most significant remaining question is regarding generalization. In the rebuttal, the authors included new results on Qwen2-VL, and showed some promising trends. However, one of the reviewers specifically asked for Qwen2.5-VL, so it is not clear that the new results would have appeased that concern. More broadly, given that there are architectural decisions that need to be made on which layers to prune, it would be more convincing to thoroughly investigate this component. The rebuttal makes a good initial step at this, but a more thorough analysis (potentially including additional architectures e.g., Qwen2.5-VL) would be needed to convincingly address this concern.

**Reviewer Scores:**

Based on the rebuttal, mo21 and 7VKh might have increased their scores to 6. The other reviewers' might have kept their scores at 4 and 2.

---

### Decision · Program_Chairs · 2026-01-26

Reject